# Trends in Incidence and Mortality of Kidney Cancer in a Northern Italian Province: An Update to 2020

**DOI:** 10.3390/biology11071048

**Published:** 2022-07-13

**Authors:** Lucia Mangone, Francesco Marinelli, Luigi Tarantini, Cristina Masini, Alessandro Navazio, Stefania Di Girolamo, Isabella Bisceglia, Carmine Pinto

**Affiliations:** 1Epidemiology Unit, Azienda Unità Sanitaria Locale-IRCCS di Reggio Emilia, Via Amendola 2, 42122 Reggio Emilia, Italy; francesco.marinelli@ausl.re.it (F.M.); isabella.bisceglia@ausl.re.it (I.B.); 2Cardiology Unit, Azienda Unità Sanitaria Locale-IRCCS di Reggio Emilia, Viale Risorgimento 80, 42123 Reggio Emilia, Italy; luigi.tarantini@ausl.re.it (L.T.); alessandro.navazio@ausl.re.it (A.N.); 3Medical Oncology Unit, Comprehensive Cancer Centre, Azienda Unità Sanitaria Locale-IRCCS di Reggio Emilia, Viale Risorgimento 80, 42123 Reggio Emilia, Italy; cristina.masini@ausl.re.it (C.M.); stefania.digirolamo@ausl.re.it (S.D.G.); carmine.pinto@ausl.re.it (C.P.)

**Keywords:** kidney cancer, trends, incidence, mortality, cancer registry, renal cell carcinoma

## Abstract

**Simple Summary:**

The aim of this work was to describe the trend of kidney tumors in a province of northern Italy through 25 years of registration. In the period examined, over 2300 patients with kidney cancer (mostly males of advanced age) and over 1200 deaths were registered, without differences between men and women but with significant age differences (12% among younger adults and 80% among the elderly). In men, we observed an increase in cases from 1996 to 2011, then the incidence decreased—probably in relation to the decline in cigarette smoking, which was also confirmed by the decline in lung cancers. Mortality decreased in both men and women, partly due to an earlier diagnosis of the disease and partly due to the availability of more advanced drugs that have made it possible to effectively treat the disease at a more advanced stage. In addition to the decrease in mortality from kidney cancer, we observed an increase in mortality from other causes, in particular from cardiovascular disease, which was also linked to the cardiotoxicity of some treatments. Therefore, along with early diagnosis and effective treatments, careful surveillance of cardiovascular episodes that may develop in these patients must be ensured.

**Abstract:**

The aim of this study was to examine the incidence and mortality trends for tumors and cardiovascular disease (CVD) in a province of northern Italy. The study included kidney cancers recorded in the period 1996–2020, divided by sex, age, year of incidence and years from diagnosis. The standardized incidence rate was calculated using the European population, and the Annual Percent Change (APC) was reported. In total, 2331 patients with kidney cancers were identified, mainly males (1504 cases) aged 60–79 years (1240 cases). There were 1257 deaths; there were no differences according sex but there were differences according to age (12.1% among younger adults and 80.4% among 80+). The incidence rate increased in males between 1996 and 2011 (APC = 2.3), while the mortality rate decreased in both males (APC = −3.3%) and females (APC = −4.5%). Comparing the same periods, kidney cancer-specific mortality decreased from 81.8% to 43.7%, while in the same period there was an increasing trend for CVD mortality. Moreover, the risk of CVD mortality increased as we moved away from the diagnosis (from 6.2% to 27.5%, *p* < 0.01). The same trend was observed for other causes of death (from 12.6% to 32.1%, *p* < 0.01). Thus, a multidisciplinary approach seems necessary during the follow-up and treatments of patients with kidney cancer.

## 1. Introduction

Kidney cancer is the ninth most frequent malignancy in Italy (with over 13,500 new cases per year), with an incidence rate of 28.1 per 100,000 in males and 11.8 per 100,000 in females [1].

The incidence in Italy appears to be increasing in both males (+2.9% per year) and females (+2.2% per year) [1]. This has also been observed in other European countries [2,3], Canada [4] and the United States [5,6]. In particular, in some countries, the increase in cases up to the 1990s was followed by a decline in incidence, linked to lifestyle changes (reduced smoking and better management of hypertension and obesity) [5].

Several risk factors have been determined to be responsible for this increase [7]. Cigarette smoking is considered a certain risk factor for kidney cancer [8,9]. Compared with non-smokers, kidney cancer among smokers is 50% higher among males and 20% higher among females [10]. Excess body weight is also a risk factor in both the US and Europe (over 40% and 30% of cases, respectively, are associated with overweight and obesity) [11]. Hypertension is responsible for 20–40% of kidney cancers [12], while environmental exposures, especially in the workplace (for example, exposure to trichlorethylene) [13], are more controversial. Moreover, a certain genetic susceptibility is responsible for a small proportion of familial forms [14], and the occasional diagnosis of abdominal masses, so-called *incidentaloma*, could be responsible for another modest increase [4,15]. In fact, the increasing use of imaging technologies (e.g., ultrasonography, computed tomography, magnetic resonance imaging) has likely resulted in greater incidental detection of kidney cancer [16], especially of smaller tumors [17,18].

At the same time, mortality has decreased in many European countries [19], in the US [20] and in Canada [6]. Early diagnosis and therapy improvements,—especially in metastatic disease with the availability of targeted therapies, immunocheckpoint inhibitors and immune-based combinations—have certainly played a fundamental role in increasing the overall survival rates in these patients [21,22,23,24].

However, the decline in mortality is not homogeneous and is strongly correlated to factors such as age and sex (mortality increases only in women and only after 75 years) [2], race (mortality falls in whites but not in American Indian/Alaska Native) [25,26] and income (a high GDP is associated with better diagnoses, treatments and survival) [20,27,28].

The strong variability of incidence and mortality recorded in various countries reflects different behaviors for the reduction of risk factors and the use of more effective treatments. However, a decrease in cause-specific mortality from kidney cancer results in an increase in survival and drug-induced cardiotoxicity which can increase mortality from cardiovascular events. Cardiac toxicity, in fact, can range from asymptomatic subclinical abnormalities, such as electrocardiographic changes and left ventricular ejection fraction decline, to life-threatening events, like congestive heart failure and acute coronary syndromes [29].

The availability of population data referring to an entire territory (and not selected from hospital cases) could allow a descriptive analysis of the incidence of new cases and the trend in mortality due to specific causes over a long time period. The aim of this study was to describe incidence and mortality trends in patients with kidney cancer diagnosed from 1996 to 2020, and to provide details on cancer and CVD mortality.

## 2. Materials and Methods

### 2.1. Data Sources

Incidence data for kidney cancer for 1996–2020 were obtained from the Reggio Emilia Cancer Registry (RE-CR). The study was approved by the provincial Ethics Committee of Reggio Emilia, Protocol no. 2014/0019740, on 4 August 2014. The main information sources of the RE-CR were anatomic pathology reports, hospital discharge records, and mortality data, integrated with laboratory tests, diagnostic reports, and information from general practitioners. The RE-CR covered a population of 529,609 inhabitants and was characterized by good data quality: kidney tumors had 89% microscopic confirmations, while DCOs (Death Certificate Only) were less than 0.1%.

Kidney cancer cases were defined based on the International Classification of Diseases for Oncology, Third Edition (ICD-O-3) [30] as topography C64.9. In total, 2331 cases of infiltrating malignant tumor (all morphologies) of the kidney incident in the Province of Reggio Emilia from 1996 to 2020 were included. The table shows the number of cases, the number of deaths and the percentages, while the age-standardized rates are shown in the figure with the trends from 1996 to 2020. In the same period, the number of total deaths from all causes in patients with kidney cancer was 1257. The causes of death were classified as overall mortality and cause-specific mortality. The latter was then divided into mortality from kidney cancer, mortality from other cancers, and mortality from CVD (heart disease, hypertension, cerebrovascular disease, atherosclerosis or aortic aneurysm/dissection; ICD-10 I00-I99).

Mortality data were selected based on classification C64 used in the International Classification of Disease, version 10 [31]. The objective of the analysis was to describe trends in incidence and mortality by age, sex and period and, for mortality, to separate cardiovascular causes of death from cancer mortality.

### 2.2. Statistical Analyses

Descriptive analyses of patient characteristics with kidney cancer were performed by number of deaths for all causes, for CVD, for kidney cancer and for other cancers. To determine the differences between these groups, we performed a one-way ANOVA test. The proportions of the causes of death were calculated by sex, age, calendar period of cancer diagnosis, and years since cancer diagnosis. We reported the trend of the proportion of CVD and malignant cancer survivors by calendar period of death.

Population estimates, which were used to derive rates, were represented by the general population of Province of Reggio Emilia recorded on January 1st of each year. Incidence rates and incidence-based mortality rates were adjusted to the 2013 European standard population and calculated per 100,000 person-years. Analyses were performed using STATA 16.1 software. In this study, we reported 95% confidence intervals (CI) and we defined a *p*-value < 0.05 as statistically significant.

Trends over time were analyzed by calculating the annual percent change (APC) in age-standardized rates using Joinpoint Regression.

## 3. Results

In the period 1996–2020, 2331 kidney cancers were diagnosed—1504 in males and 827 in females. The age-specific rate showed a net increase related to age, more marked in males than in females (Figure 1). Most cancers were diagnosed at the age of 60–79 (1240), followed by the age of 40–59 (535 cases) and then 80+ (490 cases); tumors diagnosed under the age of 40 were rare (66 cases). The number of cases between 1996–2000 was 349, and this increased to 559 in years 2011–2015, then showed a small decline of 512 in the most recent years 2016–2020. (Table 1).

The figure shows the incidence specific rates for five-year classes for males (blue line) and females (orange line).

Considering mortality, kidney cancer deaths represented the majority of cases (688 cases, equal to 54.7%), followed by 231 deaths (18.4%) from other causes, including 8 from renal failure, 183 (14.6%) from other cancers and 155 (12.3%) from CVD. (Table 1). Mortality did not differ by sex, whereas the differences were significant by age: younger adults had a significantly higher mortality from kidney cancer (*p* < 0.01), from cardiovascular disease (*p* < 0.01) and from other causes (*p* < 0.01), while there was no significant difference for other types of cancer (*p* = 0.75). Cancers diagnosed after 2010 had higher mortality from kidney cancer compared with previous years (*p* < 0.01) and reduced mortality from CVD (*p* < 0.01). Kidney cancer mortality decreased with the distance from the diagnosis (from 68.9% in the first two years to 22.1% after 10 years; *p* < 0.01), while the opposite occurred for CVD mortality (from 6.2% to 27.5%; *p* < 0.01) (Table 1 and Figure 2). Considering the year of death, in 1996–2000, the percentage of patients who died from kidney cancer was 81.8%; this dropped to 43.7% in the years 2016–2020 (*p* < 0.01), while CVD mortality rose from 6.4% to 12.3%. The increase was not significant (*p* = 0.07).

The standardized incidence rate showed an increase in males between 1996 and 2011 (APC = 2.3; 95% CI 0.6; 4.1), followed by a decline in subsequent years, although not a significant one (APC = −3; 95% CI −6; 0.2). In females, on the other hand, the trend appeared stable (APC = 0.1; 95% CI −0.7; 1) (Figure 3A). Mortality appeared to be in sharp decline in both males (APC = −3.3%; 95% CI −5.1; −1.5) and females (APC = −4.5%; 95% CI −6.2; −2.8) (Figure 3B).

The figure shows the APC (Annual Percentage Change) for males (blue line) and females (orange line). The asterisk indicates statistical significance * *p* < 0.05.

Figure 4 shows the incidence (A) and deaths (B) for the four age groups considered. The incidence, highest in the 60–79 age group, had a sharp increase until 2015 and then decreased. The trends in the 40–59 and 80+ groups were almost overlapping (Figure 4A). As regards mortality, the 60–79 age group recorded a higher absolute value in the first five years and then decreased, starting from 2006–2010, while an inverse trend was observed in those over 80 (Figure 4B).

Overall, the percentage of mortality from kidney cancer decreased over the years, going from 80% in 1996 to 41% in 2020, while mortality from CVD increased slightly, starting from 2009 at 16% (Figure 5).

The figure shows the percentage values of mortality from kidney cancer (blue line) and from cardiovascular disease (orange line), after kidney cancer diagnosis.

## 4. Discussion

The aim of our study was to examine trends in incidence and mortality for tumors and for CVD in renal cell carcinoma (RCC) patients in the Province of Reggio Emilia using data updated to 2020.

The male to female ratio was about 2 to 1, lower than that in the literature [2,5], where the former study reported a ratio of 5 to 1 and the latter 3 to 1. This derived from the fact that smoking has decreased in males in recent years, which has also been confirmed by the decrease in cases of lung cancer (Appendix A). It has been confirmed that kidney cancer is a neoplasm that affects mainly the elderly, with an increase in incidence after 40 years of age and a peak after 80 years in both males and females [32].

The incidence in males increased until 2011 (Figure 3A) and then decreased in the following period. In females, the trend appeared stable. In particular, we observed a reduction in the incidence rate that dropped since 2014, but only in males, perhaps in relation to a decrease in risk factors (smoking, in particular).

Mortality, on the other hand, decreased in both sexes (Figure 3B). In 2020, the year that coincided with the COVID-19 pandemic and consequent lockdown in Italy, there was a decline in diagnoses, more evident in males. Compared to the rest of Italy, the incidence in males in our study was lower, while in females, the figure was comparable. In Italy, the incidence trend increased, but only in the 0–49 age group, as regards males (+2.5%), and in the 50–69 age group for females (+2%) [1]. In the Province of Reggio Emilia, the trend by age was higher in the 60–79 group until 2015, after which it decreased. This trend was also confirmed by a study carried out in Denmark [2].

Incidence has increased since 2000 in males under 70 years of age [2]; Chow’s study [5] also showed an increase in Europe and North America from the 1970s to the ‘90s, which then stabilized or dropped slightly, probably due to the reduction of risk factors (smoking, obesity, hypertension). De et al. [4] also confirmed an increase in incidence from 1986 to 2007, from 13.4 to 17.9 per 100,000 in males, especially in the <65 age group, and from 7.7 to 10.3 in females. There was an increase in incidence in the United States in the period 1975–2009 [6], where the standardized rate rose from 6 to 17 per 100,000 with an APC of +2.8%. An increase was also recorded for stage I, from 4 to 12 per 100,000 with an APC value of +4.5% per year, probably due to early diagnosis. Evidence suggests that incidentally discovered renal cell carcinomas continue to constitute a major segment of all newly diagnosed renal cancers. These tumors have favorable prognoses, as they are smaller and of lower stage. On the other hand, a considerable amount of RCCs are still discovered late, after metastasis has occurred. The question was, therefore, whether early detection and hence, a screening program, would be appropriate in this setting. The available evidence did not allow for a recommendation of screening for RCC, but emphasized the importance of diagnostic biopsy of small renal tumors [33]. Even in our territory, despite the absence of a dedicated diagnostic pathway, in recent years there has been an increase in ultrasonography in subjects who request it spontaneously.

An increase in the early stages of renal cell carcinoma is associated with less invasive surgery: compared with radical nephrectomy, partial nephrectomy is associated with decreased mortality and a lower rate of postoperative decline in kidney function. Hypertension and cardiovascular events are also less frequent in conservative treatments than in demolitive ones [34]. The risk of kidney failure and worsening survival after nephrectomy could be linked, as kidney failure is itself a risk factor for cardiovascular disease and mortality [35]. Even in our series, renal failure appeared in 70% of patients over 50 and in 80% of patients over 70 undergoing nephrectomy.

The study by Li et al. [26] also confirmed an increase in the incidence in the United States among Native Americans, though much less than that observed among whites. Spain also had an increase in incidence in the years 1989–1998, mainly due to an increase in early diagnoses [3]. GLOBOCAN data [36] updated to 2012 [20,27,28] also confirmed an increase in the incidence in the United States and in northern and eastern Europe.

In northern Italy, the standardized mortality rates were 11.8 and 4.3 per 100,000 for males and females, respectively, in the years 2010–2015 [1]. In the literature, the rate has remained almost constant [2,6] or has decreased slightly, in particular in Canada, where it dropped by 0.4% per year in males and 0.8% in females in the period 1986–2009 [4]. It has decreased in the United States, but only for whites, suggesting the socioeconomic disparities present there [3,26]. Indeed, some studies have shown the extent to which GDP (Gross Domestic Product) has a negative impact on developing countries. The study by Sung et al. [20] reported a drop in mortality in the United States as opposed to an increase in the African countries. Znaor et al. [28] also reported a decline in developed countries. Azawi et al. [2] showed a decline in mortality in the 60–79 age group over the period 2001–2005. Young people with kidney cancer died from the disease, while the elderly mostly died from CVD, as confirmed by the Mangone study [37]. Kidney cancer mortality, which appeared high in the first two years after diagnosis (Figure 2), tended to decrease with the passing of the years (from 68.9% to 22.1%). The opposite appeared to hold for cardiovascular mortality, which increased gradually moving away from the date of diagnosis (from 6% to 27.5%) (Table 1). Looking at calendar years, kidney cancer patients diagnosed in 1996–2000 had nearly 82% kidney cancer mortality, which nearly halved in 2016–2020. This reduction in mortality could be linked to diagnostic anticipation and, above all, to the introduction of new drugs, especially in metastatic forms. However, the lengthening of life and the addition of cardiotoxic drugs could have contributed to the increase in CVD mortality, which rose from 6% to 12% over the years (the small numbers could explain the statistical nonsignificance). The decrease of mortality in kidney cancer by almost half in the 25 years considered could be related to better (and sometimes earlier) surgery and certainly to the introduction of effective therapies in advanced disease [38,39,40,41,42,43,44,45]. During the last decade, the treatment of advanced or metastatic kidney cancer was revolutionized with the advent of antiangiogenic drugs and tyrosine-kinase inhibitors. Several agents targeting the vascular endothelial growth factor (VEGF) pathway or the mammalian target of rapamycin pathway have been progressively approved for first-line or later-line use in the treatment of patients with advanced RCC and have become the new standard of care. As a result, the survival of patients with advanced RCC has significantly improved.

Starting from 2015, the treatment of advanced RCC underwent a second revolution with the advent of immune checkpoint inhibitors, especially agents targeting the programmed cell death-1 receptor, as well as with the advent of new generation tyrosine-kinase receptor inhibitors. More recently, the outcomes of patients affected by this once-orphan disease were further improved by the use of immune-based combinations [46].

Targeted agents have significantly improved outcomes in metastatic renal cell carcinoma and have changed long-term expectation in these patients. Although we did not have direct data on individual patient treatments, we knew that during treatments, cardiovascular adverse events have been observed, mainly in the elderly and in patients with significant comorbidities. Sunitinib has been associated with congestive heart failure (CHF) and left ventricular ejection fraction (LVEF) decline in some patients, while acute coronary syndrome has been reported with sorafenib [29]. In addition, inhibition of the VEGF receptor might be responsible for the occurrence of hypertension which may cause or exacerbate other cardiovascular diseases [47]. Regarding immune checkpoint inhibitor and novel combinations, the risk of long-term cardiotoxicity is still unknown, since most of these therapies have only been approved in the last few years. Thus, sufficient long-term safety data were not yet available.

The occurrence of cardiovascular adverse events linked to anticancer therapies could partly explain the increase in mortality related to CVD observed in the same period.

A final consideration concerned the impact that COVID-19 had on new cancer diagnoses. In 2020, the Province of Reggio Emilia was deeply affected by COVID-19. However, the urinary tract showed an increased odds ratio for hospitalization but not for death [48], while the 3-month lockdown resulted in a decrease of 35% in cancer diagnoses and the number of new kidney cancer diagnoses dropped from 117 in 2019 to 107 in 2020 [49].

One of the strengths of this study was that it involved population data referring to very recent years. A significant limitation of the study was the lack of information on the tumor stage and on treatment, which certainly had a strong impact on the prognosis of this type of tumor. However, this limit was inherent in the data collected by the Population Cancer Registers which, by definition, collect few variables but refer to an entire population and not just to a hospital case history. Additionally, the availability of data updated to 2020, which represented a limit for clinical studies, is uncommon for the population CRs, which generally have a delay of 3–4 years between incidence and publication. A more specific study, updated with data from 2021, could include other variables, such as hypertension, diabetes, comorbidities, etc., that could better explain the outcome of these patients.

## 5. Conclusions

In 25 years of registration, the kidney cancer incidence trends appeared to be decreasing in men, probably linked to the reduction of cigarette smoking. These trends appeared stable in women. Overall mortality fell in both sexes; however, ten years after diagnosis of renal cancer, mortality from cardiovascular events exceeded that of cause-specific kidney cancer.

Overall, it was determined that it will be important to monitor patients after tumor diagnosis and to provide for a multidisciplinary approach.

## Figures and Tables

**Figure 1 biology-11-01048-f001:**
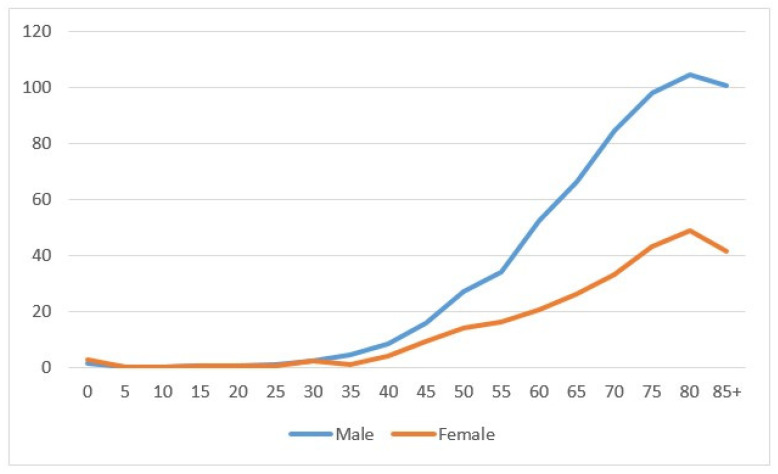
Age-specific incidence rate by sex in the Province of Reggio Emilia in the period 1996–2020.

**Figure 2 biology-11-01048-f002:**
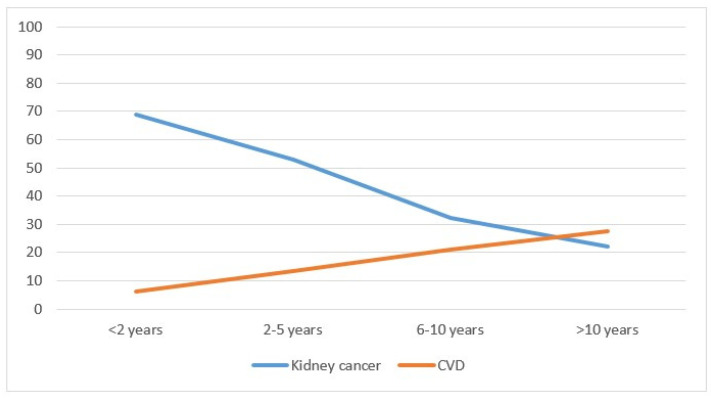
Percentage of deaths from kidney cancer and cardiovascular disease by year of cancer diagnosis in the Province of Reggio Emilia in the period 1996–2020.

**Figure 3 biology-11-01048-f003:**
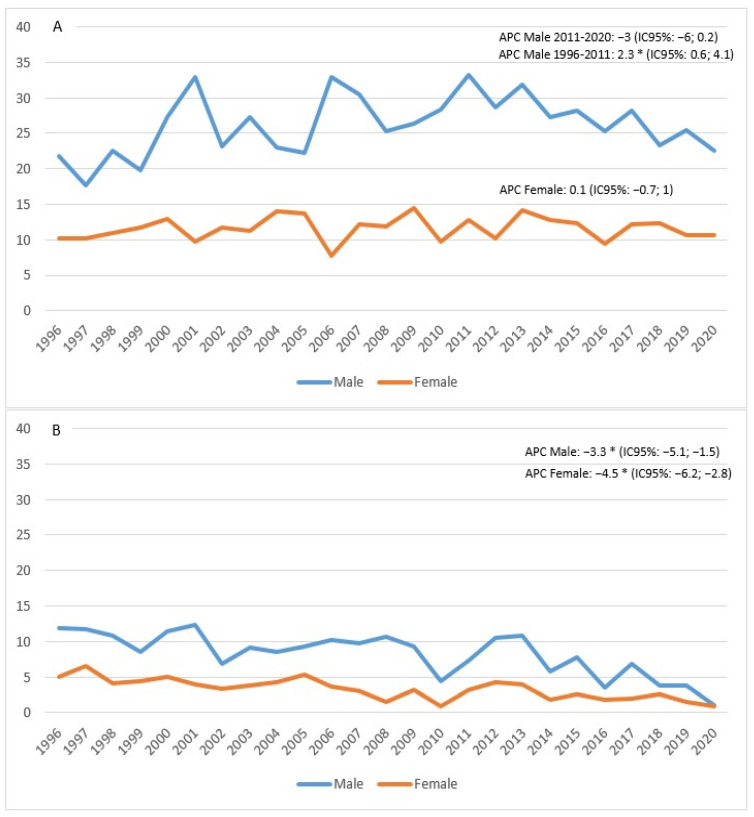
Age-standardized incidence (**A**) and mortality (**B**) rates per 100,000 p-y in the Province of Reggio Emilia in the period 1996–2020.

**Figure 4 biology-11-01048-f004:**
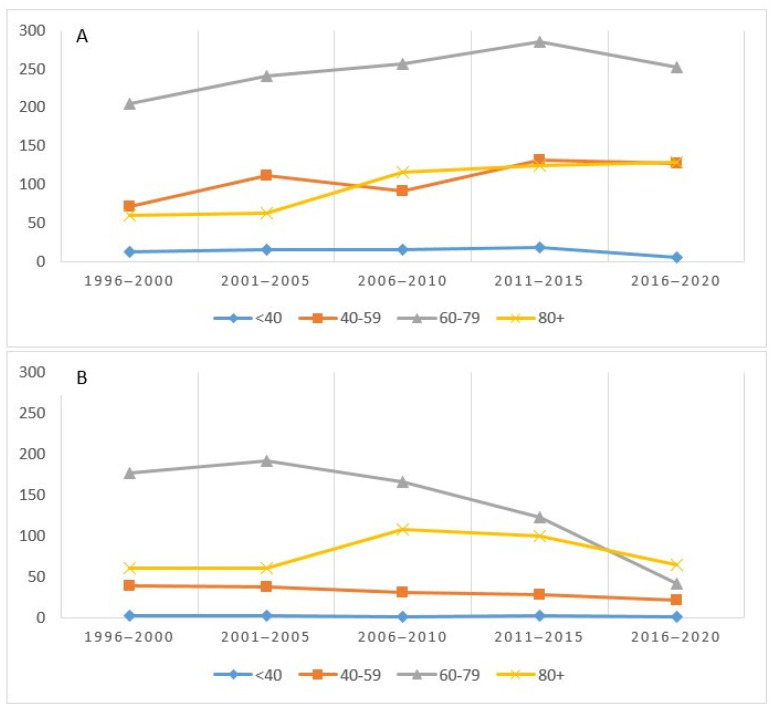
Number of cases (incidence (**A**) and mortality (**B**)) by age group and period in the Province of Reggio Emilia.

**Figure 5 biology-11-01048-f005:**
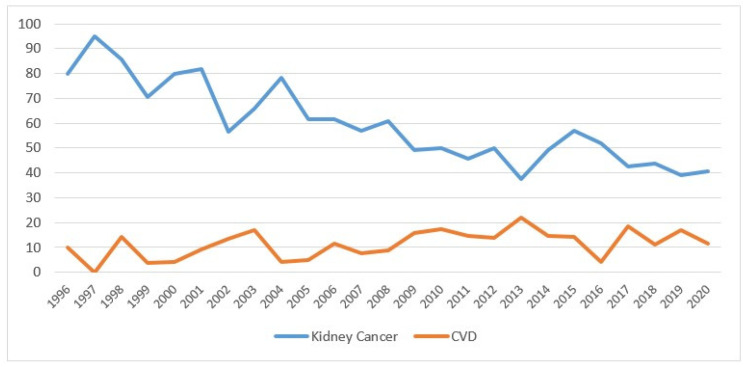
Percentage of deaths for kidney cancer and cardiovascular disease by year of death (the percentages are calculated on the total of deaths from all causes) in the Province of Reggio Emilia in the period 1996–2020.

**Table 1 biology-11-01048-t001:** Cancer patients and overall and cause-specific mortality in the Province of Reggio Emilia in the period 1996–2020.

	Incidence	Cause-Specific Mortality				Overall Deaths
	Cancer Patients	Kidney Cancer Deaths	Other Cancer Deaths	CVD Deaths	Other Causes of Death	No. of Deaths
	No.	No. (%)	No. (%)	No. (%)	No. (%)	No. (%)
**All**	2331	688 (54.7)	183 (14.6)	155 (12.3)	231 (18.4)	1257 (100)
**Sex**						
Male	1504	447 (54.9)	126 (15.5)	99 (12.1)	143 (17.5)	815 (100)
Female	827	241 (54.5)	57 (12.9)	56 (12.7)	88 (19.9)	442 (100)
*p*-value		*0.91*	*0.22*	*0.79*	*0.30*	
**Age at diagnosis**						
0–39	66	6 (75.0)	2 (25.0)	0 (0.0)	0 (0.0)	8 (100)
40–59	535	111 (70.7)	23 (14.7)	3 (1.9)	20 (12.7)	157 (100)
60–79	1240	375 (53.7)	105 (15.0)	101 (14.5)	117 (16.8)	698 (100)
80+	490	196 (49.8)	53 (13.4)	51 (12.9)	94 (23.9)	394 (100)
*p*-value		*<0.01*	*0.75*	*<0.01*	*<0.01*	
**Year of diagnosis**						
1996–2000	349	164 (59.2)	29 (10.5)	41 (14.8)	43 (15.5)	277 (100)
2001–2005	431	153 (52.4)	58 (19.9)	40 (13.7)	41 (14.0)	292 (100)
2006–2010	480	134 (43.8)	54 (17.7)	55 (18.0)	63 (20.6)	306 (100)
2011–2015	559	157 (62.1)	29 (11.5)	14 (5.5)	53 (20.9)	253 (100)
2016–2020	512	80 (62.0)	13 (10.1)	5 (3.9)	31 (24.0)	129 (100)
*p*-value		*<0.01*	*<0.01*	*<0.01*	*<0.05*	
**Years since diagnosis**						
<2 years	899	448 (68.9)	80 (12.3)	40 (6.2)	82 (12.6)	650 (100)
2–5 years	564	148 (52.9)	41 (14.6)	38 (13.6)	53 (18.9)	280 (100)
6–10 years	445	63 (32.1)	38 (19.4)	41 (20.9)	54 (27.6)	196 (100)
>10 years	423	29 (22.1)	24 (18.3)	36 (27.5)	42 (32.1)	131 (100)
*p*-value		*<0.01*	*0.05*	*<0.01*	*<0.01*	
**Year of death ***						
1996–2000	349	90 (81.8)	8 (7.3)	7 (6.4)	5 (4.5)	110 (100)
2001–2005	711	135 (69.3)	25 (12.8)	19 (9.7)	16 (8.2)	195 (100)
2006–2010	952	148 (55.6)	50 (18.8)	33 (12.4)	35 (13.2)	266 (100)
2011–2015	1161	159 (48.3)	43 (13.1)	52 (15.8)	75 (22.8)	329 (100)
2016–2020	1346	156 (43.7)	57 (16.0)	44 (12.3)	100 (28.0)	357 (100)
*p*-value		*<0.01*	*<0.05*	*0.07*	*<0.01*	

*p*-values are calculated with one-way ANOVA test. *p*-values < 0.05 are considered significant. * Percentages calculated on deaths that occurred in the period among cancer patients diagnosed from 1996, i.e., deaths in 1996–2000, only include mortality that occurred in patients diagnosed in 1996–2000, while deaths that occurred in 2001–2005 include mortality in cancer survivors diagnosed from 1996 to 2005, and deaths in 2016–2020 include mortality in cancer survivors diagnosed from 1996 to 2020.

## Data Availability

The data presented in this study are available on request from the corresponding author. The data are not publicly available due to ethical and privacy issues; requests for data must be approved by the Ethics Committee after the presentation of a study protocol.

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
