# Peer review of "Trends in Incidence and Mortality of Kidney Cancer in a Northern Italian Province: An Update to 2020"

_biology, 2022, doi:10.3390/biology11071048_

Round 1

Reviewer 1 Report

This paper aims to describe the incidence and mortality trends of kidney tumors in a province of northern Italy through 25 years. The manuscript is well written and interesting, reporting an update on the epidemiology of kidney cancer in a Northern Italian region. Nevertheless, in my opinion the paper has several flaws that make it unsuitable for publication in the current version. Hereafter my comments:

1. The aim of this study is to examine the incidence and mortality trends. However, both in tables and throughout the paper the authors often reported crude numbers, not proportions. Incidence should be reported as new cases of disease / population at start of time interval. As for mortality, throughout the manuscript it is often not so clear to the reader if the authors refer to deaths from other causes or cancer-specific deaths. For clarity, I would standardize the terminology and always refer to cancer-specific mortality and overall mortality.

2. Table 2 could be merged with Table 1. Careful to speak about incidence (again, new cases are reported, but not as a proportion). It would be clearer for the reader to have in the same table the overall deaths and cause-specific deaths. 

3. The discussion should not report again the results, as it happens in lines 192-197, and in several other parts.

4. Among limitations of study, the authors should clearly state all the limitations linked to the retrospective analysis of a cancer registry.

5. In Methods, the authors state that data were drawn from Reggio Emilia Cancer Registry: I think that the title is misleading, as the study reports data from a single Italian region only, and not all Northern Italy.

6.  The authors state that Reggio Emilia Cancer Registry data sources include anatomic pathology reports, hospital discharge records, and mortality data integrated with laboratory tests, diagnostic reports, and information from general practitioners. A Cancer Registry should be much more complete than administrative databases. Could it be possible to retrieve data concerning baseline patient characteristics (hypertension, diabetes, charlson score, previous CV status) and treatments? Without these data, the results concerning CVD mortality are purely speculative.

8. The sentence “However, the decrease in kidney cancer mortality has been associated with an increase in cardiovascular disease mortality, which increases with the disappearance of the kidney cancer diagnosis.” should be rephrased. No association can be demonstrated based on data presented by the authors.

9. The discussion is widely focused on the role of treatments for kidney cancer to decrease mortality (which is likely, but we do not know anything about treatments these patients underwent) and possibly to increase CVD (which is completely hypothetical). I believe that the authors should retrieve more data from their Registry related to possible causes of CVD. CVD could also be related to renal failure following surgical treatments from kidney cancer

10.   On line 281, the authors refer to the impact of COVID-19. The authors should discuss the fact that the major drop in uro-oncological diagnoses has followed the first COVID-19 outbreak. Can the authors retrieve the data until 2022 and not stop to 2020?

11.   DCOs: please specify what the acronym refers to.

Author Response

Many thanks for your comments! Attached is the file with the point by point answers.

Sincerely,

Lucia Mangone

Reviewer 2 Report

Title : Trends in incidence and mortality of kidney cancer in northern 2

Italian patients: an update to 2020

 Mangone et al describe in this paper the trend of kidney tumors in a province of  northern Italy through 25 years of registration.

From a statistical point of view, a good descriptive analysis has been done. In fact, the paper only aims to describe a time trend for kidney cancer and cardiovascular disease.

I would only improve the tables. Too many rows weigh down the table. Plus p-value should be written in italics

Author Response

Many thanks for your comments! Attached is the file with the point by point answer.

Sincerely,

Lucia Mangone

Reviewer 3 Report

The paper is describing the kidney cancer in northern Italy.

The data presentation and the way authors explained the problem are not appropriate and publishable in this journal.

To increase the quality of the paper I suggest to:

1. Delete northern from title 

2. The abstract and simple summary should be combined 

3. Delete numbers in the keywords

4. Introduction is too short and vague 

5. The authors must explain the relationship between kidney cancer and cardiovascular disease 

6. Conclusion must be rewritten 

Author Response

(The authors gave the same response as above.)

Round 2

Reviewer 1 Report

After proper revision, the manuscript has improved its quality.

I would recommend some revision of English language throughout the manuscript.

As for the paper, I would still focus the discussion on the main endpoint of study, which is the analysis of trends of incidence and mortality of kidney cancer. What about early diagnoses (that possibly lead to early treatment and mortality decrease?). I would also discuss the possible effect of nephrectomy on subsequent risk of renal failure and CVD. The authors could add some references on this issue. On the other hand, I would shorten the long part of discussion dedicated to systemic treatments (and their impact on CVD).

Author Response

Dear Reviewer,

thanks for the comments. We hope to have answered your requests in the best possible way.

Regards,

Lucia Mangone

Reviewer 3 Report

The introduction is too short. please explain your study procedure in the introduction briefly.

the figures are colorless. please make them bold and clear with different colors

Author Response

Dear reviewer,

thans for your comments. We hope to have answered your requests in the best possible way.

Regards,

Lucia Mangone
